# Monolithically integrated micro-supercapacitors with high areal number density produced by surface adhesive-directed electrolyte assembly

Sen Wang[1], Shuanghao Zheng[1], Xiaoyu Shi[1], Pratteek Das[1], Linmei Li[2], Yuanyuan Zhu[1], Yao Lu [2] ✉, Xinliang Feng [3,4] ✉ & Zhong-Shuai Wu [1,5,6] ✉

Accurately placing very small amounts of electrolyte on tiny micro-supercapacitors (MSCs) arrays in close proximity is a major challenge. This difficulty hinders the development of densely-compact monolithically integrated MSCs (MIMSCs). To overcome this grand challenge, we demonstrate a controllable electrolyte directed assembly strategy for precise isolation of densely-packed MSCs at micron scale, achieving scalable production of MIMSCs with ultrahigh areal number density and output voltage. We fabricate a patterned adhesive surface across MIMSCs, that induce electrolyte directed assembly on 10,000 highly adhesive MSC regions, achieving a 100 μm-scale spatial separation between each electrolyte droplet within seconds. The resultant MIMSCs achieve an areal number density of 210 cells cm$^{-2}$ and a high areal voltage of 555 V cm$^{-2}$. Further, cycling the MIMSCs at 190 V over 9000 times manifests no performance degradation. A seamlessly integrated system of ultracompact wirelessly-chargeable MIMSCs is also demonstrated to show its practicality and versatile applicability.

The interest in the Internet of Things has grown to an all-time high in today's market[1]. We are witnessing an unprecedented drive towards the microscale miniaturization of electronic devices such as implantable biosensors, micro-robots, micro-electromechanical systems, and wearable personal electronics[2,3]. However, most of the weight and volume in currently available microdevices arise from the incorporation of conventional integrated energy storage devices, which counters the original intention of micro-electronic products being lightweight and miniaturization and hinders their future development[3,4]. This highlights an urgent need for compact monolithically integrated energy storage devices with high areal number density and system performance[5,6]. Among numerous power supplies, on-chip in-plane micro-supercapacitors (MSCs) hold great potential for compact monolithically integrated energy storage devices due to their excellent and tunable electrochemical performance, superior planar geometries and compatible fabrication with on-chip integrated processing[7–9].

To date, a plethora of studies on MSCs have focused on performance modification and functional properties, yet few works focus on compact monolithically integrated MSCs (MIMSCs) with desirable

[1]State Key Laboratory of Catalysis, Dalian Institute of Chemical Physics, Chinese Academy of Sciences, 457 Zhongshan Road, Dalian 116023, China. [2]Department of Biotechnology, Dalian Institute of Chemical Physics, Chinese Academy of Sciences, 457 Zhongshan Road, Dalian 116023, China. [3]Center for Advancing Electronics Dresden (cfaed), Faculty of Chemistry and Food Chemistry, Technische Universität Dresden, Dresden 01062, Germany. [4]Max Planck Institute of Microstructure Physics, Halle (Saale) 06120, Germany. [5]Dalian National Laboratory for Clean Energy, Chinese Academy of Sciences, 457 Zhongshan Road, Dalian 116023, China. [6]University of Chinese Academy of Sciences, 19 A Yuquan Road, Shijingshan District, Beijing 100049, China. ✉e-mail: luyao@dicp.ac.cn; xinliang.feng@tu-dresden.de; wuzs@dicp.ac.cn

customizability and high areal number density in a limited space[3,10–12]. The major challenge is the lack of techniques for precise separation and localization of electrolyte on densely-packed MSCs[13], which prevents the necessary electrochemical isolation of adjoining microdevices in close proximity and interferes with other electronic components, making the compact integration of micro-electronic systems a demanding goal. A few attempts have been made to solve this intractable issue in recent years, including micro-injecting liquid electrolyte in epoxy barriers surrounding energy storage cells[14], ultraviolet curing-assisted electrohydrodynamic jet printing gel electrolyte[5], three-dimensional (3D) printing of gel electrolyte[13], and femtosecond laser-induced forward transfer technique[15]. However, these techniques typically require the accurate addition of electrolyte onto the limited project area of individual MSC one by one, and the electrolyte drops are so tiny that drop-casting without contaminating other functional electronic components is quite uncontrollable, making device assembly of MIMSCs at a large-scale time-consuming and inefficient. Dunn and coworkers designed a photo-patternable solid electrolyte as another typical attempt by chemically modifying negative photoresist using ionic liquid[16]. Isolating the solid electrolyte directly on microelectrode arrays by lithography process enables scalable manufacturing solid-state MSC with high resolution of 100 μm. However, this type of solid-state electrolyte must maintain high ionic conductivity without compromising its photopatterning functionality, so the available electrolyte options are limited. Therefore, there is an urgent need to develop an efficient, universal, and scalable technique to precisely separate and locate electrolyte droplets on densely-packed MSC arrays to produce ultra-compact MIMSCs for various on-chip integrated micro-electronic systems.

Herein, we demonstrate a universal, controllable, and customizable strategy for precise and scalable localization of tiny electrolyte droplets on large-scale microelectrode arrays with high areal number density. This is achieved by creating a patterned adhesive surface, which drives electrolyte to spontaneously anchor on 10,000 highly adhesive patterned regions with an adjacent spacing of 100 μm within seconds. Combined with high-precision lithography and spray printing to define and fabricate titanium carbide ($Ti_3C_2T_x$, MXene) microelectrodes, the resultant MXene-based MIMSCs with an inter-device spacing of 100 μm, offer an exceptional areal number density of 210 cells cm$^{-2}$ (72 cells on 0.58 cm × 0.59 cm), the highest areal output voltage of 555 V cm$^{-2}$ till date, and stable capacitance retention of 100% after 9000 cycles at an unprecedentedly-high output voltage of 190 V. Additionally, we have developed a type of seamlessly integrated ultra-compact microsystem composed of a wireless charging coil and MIMSCs (WC-MIMSCs) by taking advantage of the flexible and versatile connection mode of MSCs. The WC-MIMSCs efficiently power a display screen for 30 min with wireless charging of only 2 s, demonstrating the great potential of MIMSCs in practical applications.

## Results

### Microfabrication of compact and customizable MIMSCs
The on-chip micromanufacturing of MIMSCs is illustrated in Fig. 1 and Supplementary Fig. 1. Microelectrode arrays of MIMSCs were first obtained by multi-step lithographic patterning, magnetron sputtering, spray printing aqueous dispersion of small-sized MXene nanosheets (Supplementary Fig. 2) and lift-off processes (Supplementary Fig. 1a)[13]. By this way, 10,000 integrated microelectrode arrays with a fine characteristic size of 50 μm, an ultra-small projected area of 0.36 mm$^2$, and an inter-cell spacing of 100 μm, were successfully produced on a 6.5 cm × 7.5 cm substrate (Fig. 2a–c).

Adherence of electrolyte to individual cell area and preventing contact with the electrical interconnects and adjacent cells (*i.e.*, ensuring the electrochemical isolation of each microcell) are of great importance for integrated energy storage devices. Figure 1 and Supplementary Fig. 1b depict our approach for superfast and precise

deposition of electrolyte induced by patterned adhesive surface. First, microelectrode arrays (the region 1 in Fig. 1b), including electrode and electrode interspace, with high adhesion to electrolyte (Fig. 1c, d), were protected with a layer of photoresist (Supplementary Fig. 3) after a general lithography process to prevent any modification by subsequent processing. Second, the whole surface was modified with 1H, 1H, 2H, and 2H-perfluorodecyltrimethoxysilane (FAS-17) to greatly reduce its adhesion (Fig. 1e). Subsequently, the patterned adhesive surface was obtained by washing away the photoresist layer using ethanol. Time-of-flight secondary ion mass spectrometry was performed to analyze the distribution of FAS-17. The top view image of -F fragment represents the distribution of FAS-17 (Supplementary Fig. 4). It is clearly seen that FAS-17 has a very uniform distribution over a large area of 3.5 mm × 6.1 mm, which is consistent with the designed pattern before lift-off process of photoresist in ethanol, indicating good stability of FAS-17 patterns during the process. Additionally, the stability and consistency of FAS-17 layer on glass during the photoresist lift-off process was also verified by the unchanged static contact angle (SCA) at different positions during immersion in ethanol for 10 cycles (Supplementary Fig. 5). Ultimately, benefiting from the staggering difference between the respective surface adhesion, when $EMImBF_4$ electrolyte solution flowed over the patterned substrate, it spontaneously anchored on 10,000 highly adhesive regions, achieving spatial separation of electrolyte within seconds, as observed in Figs. 1b, 2d, and Supplementary Video 1. Additionally, $EMImBF_4$ electrolyte could spontaneously directed assembly to different shapes depending on the customizable geometries of microcell arrays (Fig. 2e), thus achieving electrochemical isolation of aesthetically diverse cells (Fig. 2f). Furthermore, aqueous, ionic liquid, and organic electrolytes have similar wetting behavior with the patterned adhesive surfaces, forming stable and consistent droplet arrays with clear profiles (Fig. 2g) and achieving electrochemical isolation of microelectrode arrays in different electrolytes. Also, this strategy can be easily adapted to deposit a functional material such as MXene solution on the patterned area to prepare microelectrodes (Fig. 2g), indicative the broad applicability of our approach.

To understand the mechanism of such precise electrolyte assembly over the patterned adhesive surface, we compared the surface adhesion properties of $EMImBF_4$ electrolyte over the FAS-17-treated and untreated regions. As shown in Fig. 1c–e, the SCAs between $EMImBF_4$ electrolyte and MIMSCs region (labeled region 1 in Fig. 1b) constituting MXene microelectrodes and interdigital spaces are 83° and 63°, respectively, suggesting greater wettability compared to the FAS-17-treated region (between adjacent cells, labeled region 2 in Fig. 1b) with the SCA of 98°. However, the major driving force behind the patterned distribution of the electrolyte is the large difference between the adhesivity characteristics of region 1 and 2. The receding contact angle (RCA) between $EMImBF_4$ electrolyte and region 1 was less than 5°, indicating high adhesion[17,18]. This explains why electrolyte droplet contact with the MXene film or untreated glass surface strongly adheres to them and is difficult to remove (Supplementary Videos 2–4). However, the RCA was more than 120° for FAS-17-treated glass indicating its low adhesion[17,18]. Naturally, the electrolyte droplet slips easily without leaving residue due to the substantially low adhesion (Supplementary Video 5). Consequently, when the electrolyte flows over a patterned adhesive surface with large difference in adhesion, it spontaneously gets anchored on the more adhesive region and slides off the less adhesive region (Supplementary Videos 6 and 7), thus achieving precise localization of the electrolyte in the desired area. Additionally, the treated MIMSCs also demonstrated similar wetting behavior with water as shown in Supplementary Fig. 6.

### Electrochemical performance of the individual MSC
To highlight the superior electrochemical performance of our MSCs, we first investigated the cell with an ultra-small projected area of

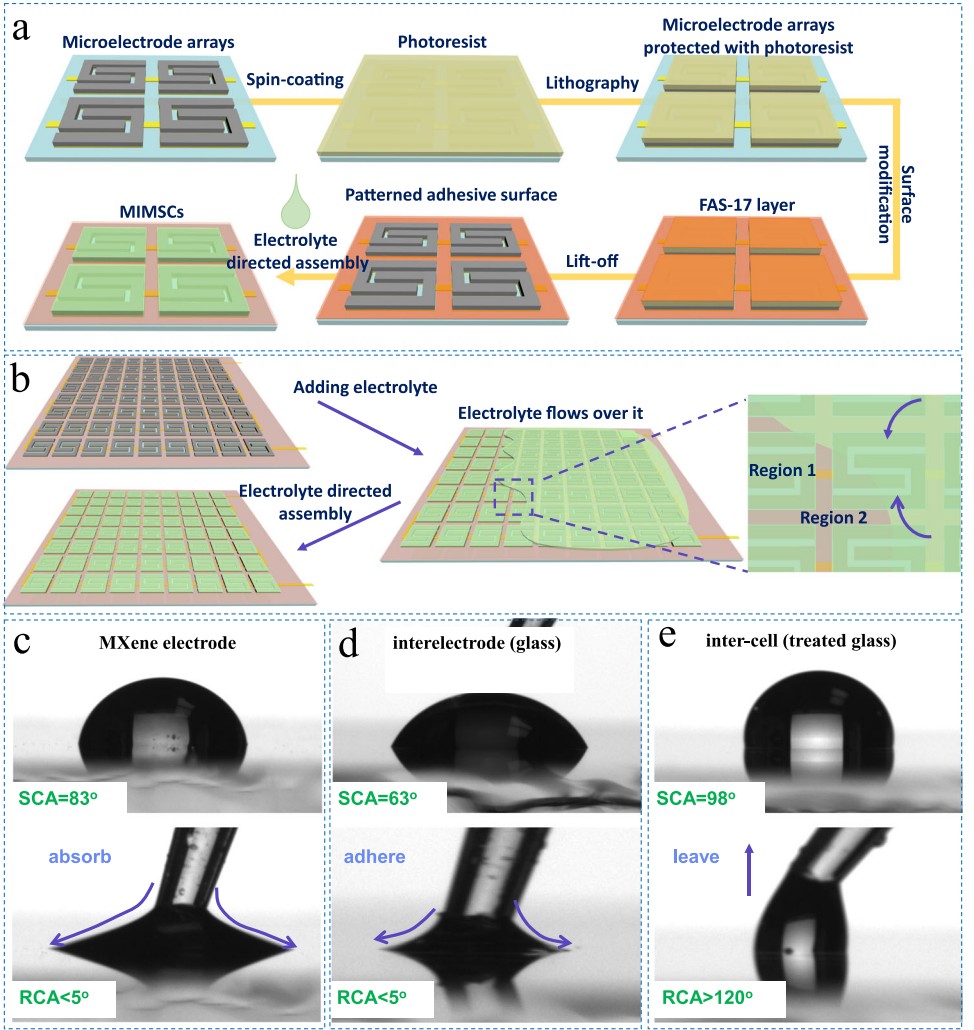

**Fig. 1 | Technique and mechanism for microfabricating MIMSCs. a** Schematic of the MIMSCs microfabrication process. **b** Mechanism of patterned adhesive surface inducing electrolyte-directed assembly. **c**–**e** The SCAs and RCAs of EMImBF$_4$ electrolyte with MXene microelectrodes (**c**), interelectrode glass (**d**), and treated inter-cell glass (**e**).

0.36 mm² (Fig. 3a, microelectrode thickness of 250 nm) in an ionic liquid EMIMBF$_4$ electrolyte. As expected, the galvanostatic charge-discharge (GCD, Fig. 3b, c) profiles at different current densities were nearly symmetrical triangular shapes, indicating the typical capacitive behavior of MXene nanosheets. Our MSC presented an areal capacitance of 2.3 mF cm$^{-2}$ at 0.15 mA cm$^{-2}$ (Fig. 3d) as a result of the superior electrochemical activity of small-sized MXene nanosheets arising from abundant edge planes and favorable electron-ion transport paths. Moreover, the MSC also displayed excellent rate capability under a high current density of 3.85 mA cm$^{-2}$ with an appreciable capacitance of 2.06 mF cm$^{-2}$. The coulombic efficiency almost reached 100% at both small and large current densities from 0.15 to 3.85 mA cm$^{-2}$ (Fig. 3d) with minimal leakage current. Meanwhile, the capacitance retention was close to 100 % after 10,000 cycles (Fig. 3e). To adjust the operating voltage and output capacitance of an individual MSC, we further evaluated its electrochemical performance in 1 mol L$^{-1}$ H$_2$SO$_4$ electrolyte and 20 mol kg$^{-1}$ LiCl electrolyte. As observed from GCD profiles (Fig. 3f and Supplementary Figs. 7 and 8), our MSC delivered a voltage window of 0.6 V in 1 mol L$^{-1}$ H$_2$SO$_4$ electrolyte, and 1.6 V in 20 mol kg$^{-1}$ LiCl electrolyte, respectively. Thereby, the MSCs delivered a considerable areal capacitance of 9.3 mF cm$^{-2}$ in 1 mol L$^{-1}$ H$_2$SO$_4$ electrolyte, and 6.0 mF cm$^{-2}$ in 20 mol kg$^{-1}$ LiCl electrolyte at 0.23 mA cm$^{-2}$, respectively (Fig. 3g). It is calculated that the MSCs exhibited an energy density of 15.7 mWh cm$^{-3}$ (0.42 µWh cm$^{-2}$)

in EMIMBF$_4$, 15.3 mWh cm$^{-3}$ (0.38 µWh cm$^{-2}$) in 20 mol kg$^{-1}$ LiCl, and 3.3 mWh cm$^{-3}$ (0.08 µWh cm$^{-2}$) in 1 mol L$^{-1}$ H$_2$SO$_4$; comparable with previously reported MXene-based MSCs[19–25].

## MIMSCs with unprecedented areal number density and areal output voltage

Besides employing electrolytes to regulate the output voltage/capacitance, another more straightforward approach is to integrate multiple MSC units in series and/or parallel, thus generating a superposition of output voltage/capacitance[26]. So far, constructing MIMSCs with high-density and stable performance has remained challenging. To demonstrate the advantages of our strategy, MIMSCs comprising 72 cells with customized serial and/or parallel configurations, were integrated on a small substrate (0.58 cm × 0.59 cm, Fig. 4a and Supplementary Fig. 12a). The interelectrode spacing of the microcell was 50 µm, and the inter-cell spacing was only 100 µm, achieving a high space utilization and reaching an excellent areal number density of 210 cells cm$^{-2}$, far superior to the previously reported integrated MSCs, with number density lower than 158 cells cm$^{-2}$ (Fig. 4h)[27–30]. The successful ability to compactly integrate fully functioning MSCs is attributed to the spatial control of electrolyte (Supplementary Fig. 9) by constructing patterned adhesive surface on the entire MIMSCs areas. Moreover, even a slight discrepancy between individual microcells may lead to a big difference in voltage division, cascading into uneven

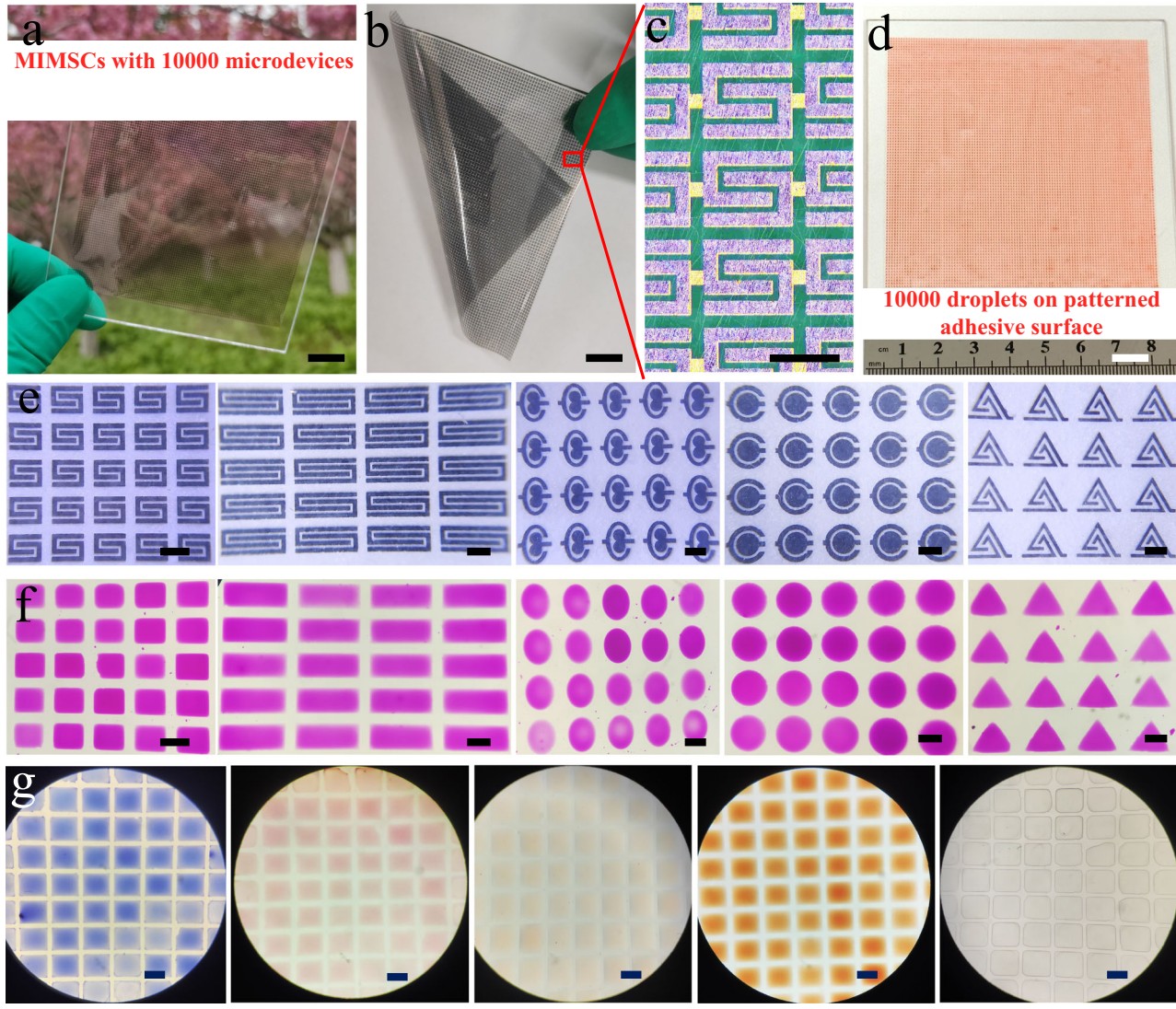

**Fig. 2 | Characterization of MIMSCs. a**, **b** Photograph of MIMSCs containing 10,000 MSCs on a wafer scale glass substrate (**a**), and a flexible polyethylene terephthalate substrate (**b**). **c** Microscope image of MIMSCs. **d** Photograph of organic solvent stained by Sudan II assembly on 10,000 square arrays on the patterned adhesive substrate. **e** Photographs of MIMSCs with various customizable geometry configurations (from left to right: square, rectangle, elliptic, circular, and triangle).

**f** KOH electrolyte stained with phenolphthalein anchored on arrays of various shapes (from left to right: square, rectangle, elliptic, circular, and triangle). **g** Different solutions anchored on rectangles (from left to right: sulfuric acid stained by methylene blue, sodium sulfate stained by neutral red, EMIMBF$_4$ stained by Sudan II, dimethyl carbonate stained by methyl orange, and MXene solution). Scale bars in (**a**, **b**, **d**) are 1 cm, and in (**c**, **e**–**g**) are 500 μm.

energy distribution throughout the monolithic system. This will result in some units not being fully activated while others are overworked, eventually causing the MIMSCs to fail unpredictably. Thus, the performance consistency among all the microcells is imperative to the successful integration. The 3D thickness mapping of a random region and the height profiles of eight individual MSCs in the MIMSCs showing the consistency of microcells' thickness, indicate that the microfabrication process was highly controllable over a large scale (Fig. 4b, c and Supplementary Fig. 10). Benefiting from the thickness consistency and accurate coverage of electrolyte on microcells, the capacitance of all the microcells varies from 1.55 to 1.77 μF (Fig. 4d), which ensures the monotonously increasing voltage to 190 V of MIMSCs with 72 serially-connected microcells-in EMIMBF$_4$ electrolyte (Fig. 4e). Additionally, the correlation coefficient calculated from the plot of response capacitance versus cell number was close to 1, demonstrating the excellent modularity and performance uniformity of all the on-chip MIMSCs (Fig. 4e). Besides, our MIMSCs could operate under 190 V stably with acceptable coulombic efficiency at different

current densities (Fig. 4f), ultimately delivering a high systemic energy density of 14.8 mWh cm$^{-3}$ (0.37 μWh cm$^{-2}$) based on the whole model including inter-cell spacing. The self-discharge time-of fully charged MIMSCs from 190 V to 82 V is 1.5 h (Supplementary Fig. 11) because the reliable electrochemical isolation effectively prevented the leakage current between individual microcells. This pack of MIMSCs demonstrated almost 100% capacitance retention after 9000 continuous long-term cycles at an applied voltage of 190 V (Fig. 4g), demonstrative of exceptional performance consistency among all the microcells. It should be noted that the value of 190 V is one of the highest operating voltages ever reported[28,30–33]. The corresponding areal voltage of 555 V cm$^{-2}$ was the highest value for serially integrated MSCs to date, outperforming all previous reports (Fig. 4h), in which electrolytes were evenly divided using different techniques, such as manually adding electrolyte for inkjet-printed MXene/PH1000 MSCs (5.4 V cm$^{-2}$)[27], laser carving of gel electrolyte for graphene MSCs (18.52 V cm$^{-2}$)[33], ultra-violet curing-assisted electrohydrodynamic jet printing of gel electrolyte for nano activated carbon MSCs (65.9 V cm$^{-2}$)[5], 3D printing gel

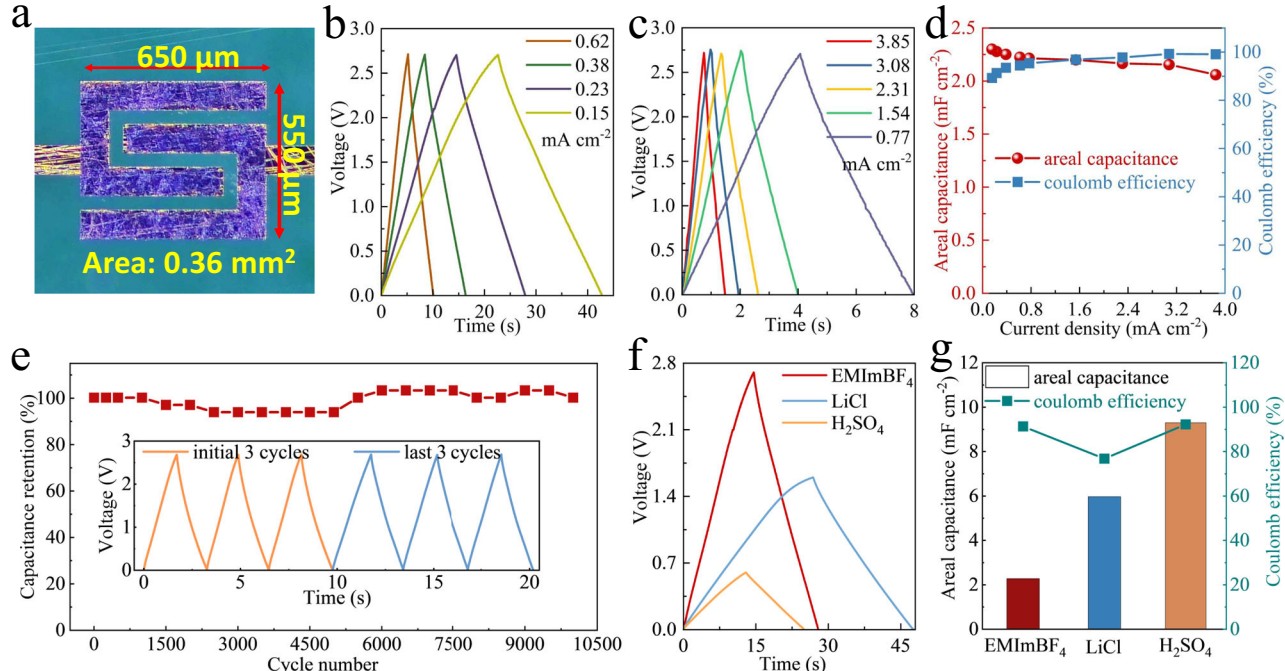

**Fig. 3 | Electrochemical performances of an individual MSC in different electrolytes. a** Optical microscope photograph of an as-fabricated MSC with an area of 0.36 mm². **b**–**e** GCD profiles over current densities from 0.15 to 3.85 mA cm⁻² (**b**, **c**), areal capacitance and coulombic efficiency as a function of current density (**d**), and cycling stability (**e**) of MSC in EMIMBF₄ electrolyte. **f**, **g** GCD profiles (**f**), areal capacitance, and coulombic efficiency (**g**) of MSC in EMIMBF₄, 20 mol kg⁻¹ LiCl, and 1 mol L⁻¹ H₂SO₄ electrolytes measured at 0.23 mA cm⁻². Source data are provided as a Source Data file.

electrolyte for MXene MSCs (75.6 V cm⁻²)[13], cutting out and surface-mounting for separating polyaniline MSCs (80 V cm⁻²)[6], and micro-injecting liquid electrolyte for carbon nanotube MSCs (158.7 V cm⁻²)[14]. Meanwhile, the total current/capacitance response can be easily enhanced by an in-parallel configuration of customizable MIMSCs. As observed in Supplementary Fig. 12, the simultaneous increase in overall voltage, current, and capacitance of the highly complex modular MIMSCs, arranged as multiple parallel rows (1-9) of 8 serially-connected microcells, also indicated the ultra-compactly integration and performance uniformity. Therefore, our strategy is highly promising for producing monolithically integrated microscale power supplies to satisfy the varying customization demands in actual scenarios. For instance, given that our MIMSCs assembled on a flexible substrate weighed only 1 mg (Supplementary Fig. 13), it could prove to be critical for delicate micro-robots which cannot carry heavy energy-supplying components[34].

We also produced MIMSCs with higher number density per unit area by fixing the inter-cell spacing to 100 μm, changing the microcell width from 500 to 200 μm, with the aim of exploring the precision limitation of our strategy. As shown in Supplementary Fig. 14, micro-electrode arrays with microcell width up to 200 μm were produced and electrolytes were successfully anchored onto microcell arrays by our strategy. For example, the 3 × 3 MIMMSCs, in which each MSC unit has an electrode length of 200 μm, width of 75 μm, gap of 50 μm and an inter-cell spacing of 100 μm, achieved an ultra-high areal cell number of 1404 cell cm⁻².

**Seamlessly integrated wireless charging MIMSC microsystem**
Replacing the traditional electric supply component with non-contact charging can improve the practicality of the energy storage micro-devices in implanted electronics, micro-drones, and micro-detection systems by achieving high integration while eliminating the cumbersome procedure of externally connecting the circuits[35–37]. To this end, we developed an integrated wireless charging energy storage

microsystem composed of a wireless charging coil and MIMSCs (WC-MIMSCs, Fig. 5a, b). As shown in Supplementary Fig. 15, the WC-MIMSCs were designed with a small size of 1.19 cm × 1.1 cm, in which the coil was in the center surrounded by 60 MSCs, sharing an electrode (orange line in Supplementary Fig. 15). The seamless coupling of WC-MIMSCs was realized by simply changing the pattern of the lithographic mask during the current collector fabrication process of MIMSCs (Supplementary Fig. 16). The output voltage of our MIMSCs, arranged as 30 in-parallel rows of 2 serially-connected microcells, was doubled, and the output capacitance was raised by nearly 15 times compared to a single MSC, suggesting a superior serial and parallel behavior of the fabricated MIMSCs (Supplementary Fig. 17).

A typical wireless charging system consists of a transmitter and a receiver, and its working principle is demonstrated in Fig. 5a, b. As shown in Fig. 5c, the voltage of the MIMSCs rose to 5.4 V in seconds when the receiving coil had mutual inductance with transmitting coil charged by a 10 V direct current power supply. Upon removing the transmitting coil, the MIMSCs could steadily discharge over a long duration at 1 μA (Fig. 5d). According to the discharge profiles and capacitance of MIMSCs obtained after wireless charging for different periods, it can be seen that only wireless charging of 2 s, the MIMSCs achieved 88% charging capacitance. Meanwhile, the voltage reached 5.4 V, and the current of wireless charging rapidly decreased[35,36]. The MIMSCs continued to be charged under a low current and the capacitance maintained almost unchanged after wireless charging of 20 s, validating the applicability of our MIMSCs.

It is notable that after wireless charging of 2 s, the as-transmitted energy stored in the MIMSCs module could drive the operation of a display screen for 30 min, showing its great potential for actual applications (Fig. 5e and Supplementary Fig. 18).

## Discussion
In summary, we have successfully tackled a major obstacle in creating ultra-dense MSCs by employing a new method. This method

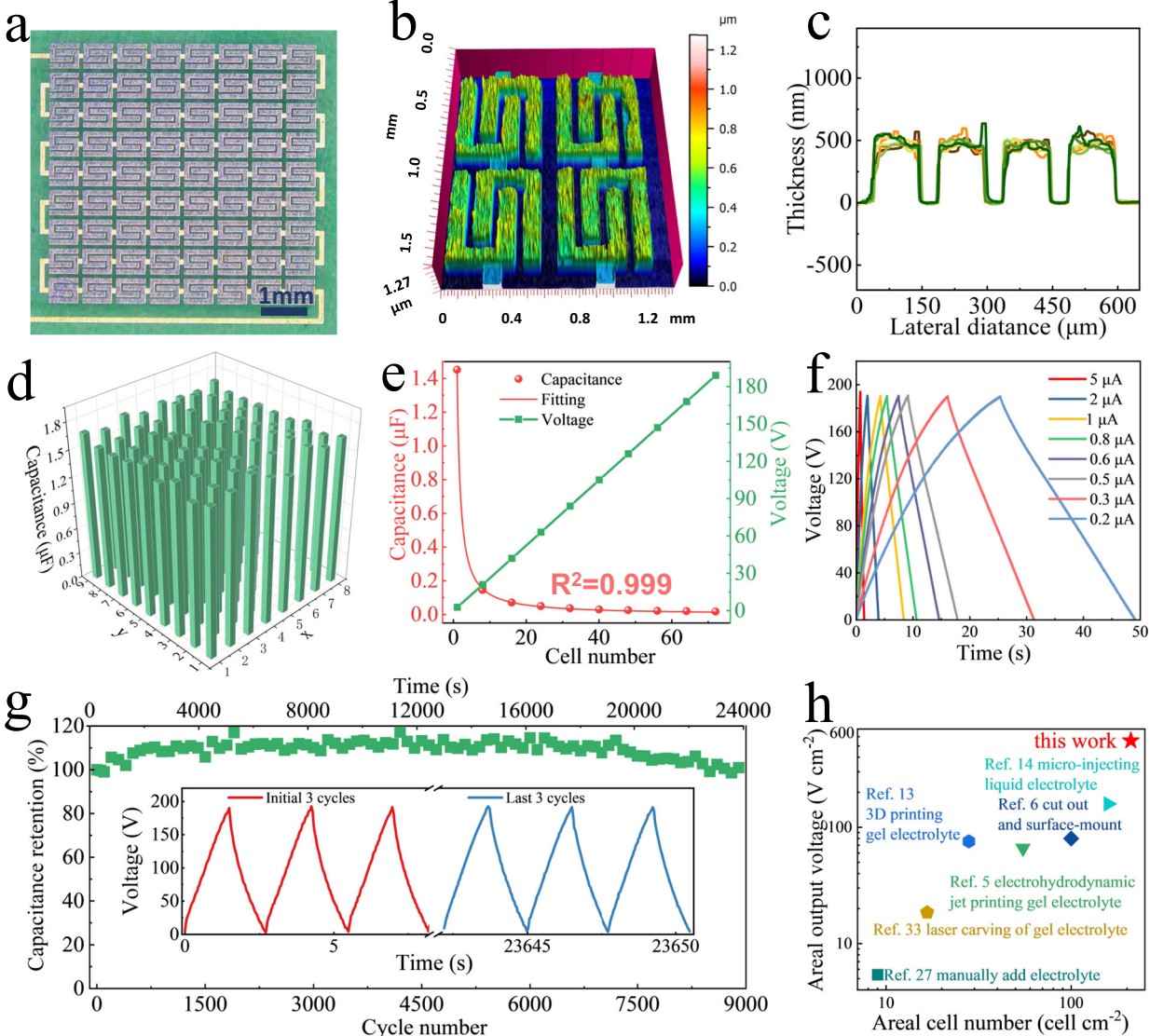

**Fig. 4 | Electrochemcial characterization of MIMSCs with 72 serially-connected microcells. a** Photograph. **b, c** 3D thickness mapping of four selected MSCs (**b**) and height profiles of randomly selected MSCs (**c**) at different locations in MIMSCs model. **d** Capacitance obtained at 1 μA of each MSC unit within the MIMSCs. **e** Total voltage/capacitance response versus cell number of MIMSCs containing 8, 16, 24, 32, 40, 56, and 72 serially-connected microcells, the capacitance was calculated from GCD profiles measured at 2 μA. **f, g** GCD profiles obtained at currents of 0.2−5 μA (**f**) and long-term cycling stability (**g**) of 72 serially-connected microcells under 190 V. Inset in (**g**) shows the initial and last 3 GCD profiles during the 9000-cycle run. **h** Areal cell number and areal output voltage of our MIMSCs compared with previous reports[5,6,13,14,27,33]. Source data are provided as a Source Data file.

involves a patterned adhesive surface that induces direct assembly of electrolyte into 10,000 tiny droplets within seconds. The on-chip MIMSCs attained the highest areal cell number density (210 cells cm⁻²) and areal voltage of 555 V cm⁻² reported to date, while remarkably retaining 100% of the initial capacitance after 9000 cycles at a tremendously high output voltage of 190 V. The wirelessly-chargeable energy storage microdevice could provide an uncompromised performance, demonstrative its practicality and versatile applicability. The dense integration of the microcells is a culmination of the high-resolution microfabrication techniques for the micro-electrode arrays and precise electrolyte droplet-directed assembly induced by a patterned adhesive surface. Moreover, our strategy is highly flexible and applicable to other integrated MSCs and micro-batteries based on different electrode materials and electrolytes, holding great promise for achieving high modular output performance requirements for compactly integrated Internet of Things technologies.

## Methods

### Synthesis of Ti₃C₂Tₓ MXene

Ti₃C₂Tₓ MXene was synthesized by selective etching of Ti₃AlC₂ using LiF/HCl mixture[13]. Firstly, 0.5 g LiF was dissolved in 10 mL HCl (9 mol L⁻¹) solution with constant stirring to form etchant. Secondly, 0.5 g Ti₃AlC₂ was added to the etchant solution slowly to avoid over-heat and etched at 35 °C for 24 h. Thirdly, the etch product was washed using deionized water combined with centrifuged at 3500 rpm for many times until pH reached 6. Then, the resulting Ti₃C₂Tₓ precipitate was mixed with 10 mL deionized water, followed by ultrasonic 30 min in an ice water bath to delaminate the multilayer Ti₃C₂Tₓ into single layer. Subsequently, the dispersion was centrifuged at 1500 rpm for 1 h to remove the unetched Ti₃AlC₂ or multilayer Ti₃C₂Tₓ, and then the collected upper layer solution was subjected to further high-speed centrifugation (3500 rpm, 1 h). Finally, to obtain smaller flake sizes for further usage, MXene suspension was diluted to 1 mg mL⁻¹ and then sonicated for 15 min using a tip sonicator while stirring in an ice bath.

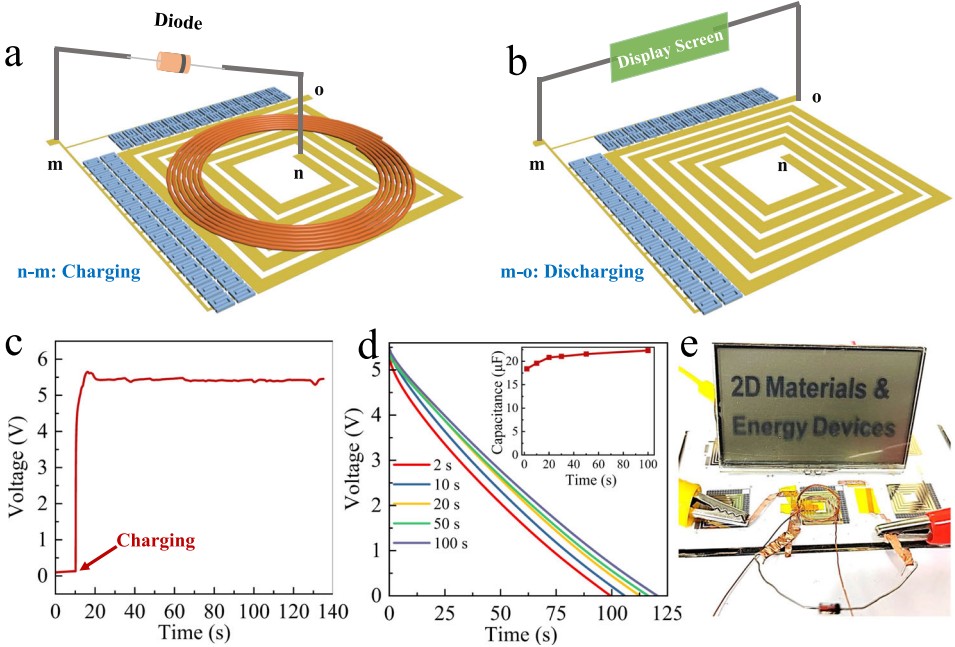

**Fig. 5 | Characteristics of WC-MIMSCs microsystem. a, b** Schematic diagram of charging and discharging. **c** Charging curve of MIMSCs charged by a wireless coil. **d** Discharge curves at 1 μA of MIMSCs after charging at 5.4 V for different times (2–100 s), inset is corresponding capacitance. **e** Photograph of a display screen powered by WC-MIMSCs. Source data are provided as a Source Data file.

## Fabrication of MIMSCs microelectrode arrays

First, the substrates (e.g., Si, glass, and flexible polyethylene terephthalate) were cleaned by sequential bath in ethanol and deionized water for 1 h. Next, a thin photoresist (AZ4620) was coated on the target substrate and then exposed to ultraviolet light through a photomask, forming the pre-designed patterns of metal current collectors. After development in AZMIF-300 solution, the exposed regions of photoresist were dissolved, leaving behind the pre-designed pattern. Next, a thin layer of Au/Ti was sputtered over the substrate followed by lift-off procedure enabled by immersion in acetone, forming the current collectors for electrodes, electrical connections between adjoining microcells, and for performing external measurements. Interdigitated fingers have a typical width of 100 μm, length of 650 μm, interspace of 50 μm, and cell-to-cell spacing of 100 μm. Then, another photoresist layer (S1805) with microelectrode pattern on Au/Ti metal collectors was obtained by the same process. Finally, MXene microelectrodes were achieved by spray printing 1 mg mL$^{-1}$ MXene dispersion by an automatic spraying equipment, followed by lift-off in acetone assisted by ultrasonic treatment.

## Electrolyte-directed assembly strategy

To guarantee the electrochemical isolation of adjoining microcells in close proximity, electrolyte was precisely separated and localized on microcells by an electrolyte-directed assembly strategy on patterned adhesive surface. As shown in Fig. S1b, after a general lithography process, a rectangular lithography layer (AZ4620), with a typical width of 560 μm and length of 660 μm, slightly larger than a microelectrode, was left on the surface of the microelectrode arrays. Next, the surface was modified with FAS-17 by chemical vapor deposition at 70 °C for 3 h to reduce the adhesion to the electrolyte. Subsequently, the patterned adhesive surface was obtained by washing away the photoresist covering layer with ethanol. Finally, when the electrolyte solution was flushed on the patterned substrate, benefiting from the large disparity between surface adhesion in the different regions, it spontaneously anchored on the microelectrode array regions, instantly.

## Reporting summary

Further information on research design is available in the Nature Portfolio Reporting Summary linked to this article.

## Data availability

The data generated in this study are provided in the main text and Supplementary information, where the source data of Figs. 3, 4, and 5 are listed in the Source Data file. Extra data are available from the corresponding author upon reasonable request. Source data are provided with this paper.

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

## Acknowledgements

This work was supported by the National Natural Science Foundation of China (22125903, 22209175, 22075279, 22279137, 22005297, 21805273, 22109160, and 21804133), the National Key R&D Program of China (Grant 2022YFA1504100, 2023YFB4005204), the Dalian Innovation Support Plan for High-Level Talents (2019RT09), the Dalian National Laboratory For Clean Energy (DNL) Cooperation Fund, the Chinese Academy of Sciences (DNL202016 and DNL202019), Dalian Institute of Chemical Physics (DICP) (DICP I2020032), and China Postdoctoral Science Foundation (2021M693127).

## Author contributions

Z.-S.W., X.F. and Y.L. conceived the experiments and supervised this project. S.W. performed the preparation, characterization, and performance measurement of all the devices. S.Z. carried out the fabrication and characterization of MXene nanosheets. X.S. carried out the construction of wireless charging microsystem. P.D. and Y.Z. analyzed the electrochemical data. L.L. carried out the lithographic patterning process. S.W., P.D., Y.L., X.F. and Z.-S.W. wrote the manuscript. All the authors discussed the results and commented on the manuscript.

## Competing interests

The authors declare no competing interests.
