## [Peer Review File · Nature Communications]

Monolithically integrated micro-supercapacitors with high areal number density produced by surface adhesive-directed electrolyte assemblyREVIEWER COMMENTS

Reviewer #1 (Remarks to the Author):

This manuscript describes a self-splitting mechanism of aqueous electrolytes, enabling the fabrication of compact and customizable micro-supercapacitors. The method of preparing the devices is novel, relying on a unique FAS-17 treated glass surface to tune adhesion properties. This allows precise electrochemical isolation between devices. This approach avoids coating the entire surface, thereby minimizing the risk of contamination to other functional electronic components. The devices are comprehensively characterized for their properties and performance, with particular emphasis on the ability to maintain an exceptionally high areal output voltage of 555 V cm⁻² and a stable capacitance retention of 100% after 9000 cycles, even at a high output voltage of 190 V (fig 4). Moreover, the device exhibits wireless charging capabilities with receiving and transmitting coil system, which could drive a display screen for 30 minutes. This represents a significant advancement in potential microelectronics applications. In summary, the manuscript presents a novel device with significant potential for various microdevices, showcasing precise control of electrolyte localization. However, prior to recommending it for publication, the following questions need to be addressed and discussed.

The authors claim to pattern the electrolyte by a spontaneous droplet self-splitting strategy on a treated surface. However, it needs further intervention by sucking up the excess solution using a pipette to trigger manual or partially guided self-splitting of electrolytes (Supplementary Video 7).

1. It's a bit premature to claim it as a self-splitting strategy please comment on this. How can the authors precisely control the amount or volume of electrolyte that is removed despite strong hydrogen bonding between the electrolyte droplets? Due to the same reason how do the authors guarantee electrochemical isolation between devices, particularly given the narrow inter-device spacing of 100 μm?
2. Moreover, the authors investigated the surface's adhesion of electrolyte droplet (fig 1c). However, it is important to consider the possibility of electrolyte contact between devices if the entire system is tilted or moved creating electrochemical short. Comments addressing this concern would be helpful.
3. One aspect that is unclear is the consistency and stability of FAS-17 material patterning, especially during lift-off process of photoresist, without encountering issues of delamination or peeling-off the FAS-17 layer (fig 1a)?
4. The authors charged 72 cells connected in series (arranged in 8 rows) and parallel (9 columns) at 190 V. Consequently, each individual device is minimally charged at over 20 V. This voltage not only raises concerns about high voltage arcing but also may cause gas evolution. It is imperative for the authors to elucidate how the device addresses this issue. Given that the stability window in aqueous environment is only 1.21 V, and with an air break done voltage ~ 3V/μm and the authors charged the individual MSC only up to 2.5 V (instead of the 20V claimed) (fig 3) in three different ionic liquid-based electrolytes (EMIMBF₄, LiCl and H₂SO₄). Charging the MSC beyond the stability window may cause water splitting reactions and damage the device. The authors should provide information on the stability of device when operating beyond the range of electrochemical water splitting.
5. In considering practical applications beyond wireless charging system, the intrinsic performance of self-discharge is crucial, defining the duration for which the system can function independently without requiring an external charge. The authors charged the array of MIMSCs at a voltage of 190 V, and it would be interesting to investigate the self-discharge behavior after it charges up to 190 V. given such high voltage one would expect strong side reaction, material degradation, water splitting and arc discharge.

Reviewer #2 (Remarks to the Author):

This is a very interesting report on a novel technique to fabricate micro-supercapacitors in a way that allows larger scale series connections of devices on a smaller footprint area than what has been shown before. The authors convincingly demonstrate the viability of the technique and the performance, in particular in terms of generating a high voltage per footprint area, is impressive.

I appreciate the effort to show performance of the devices using a range of electrolytes. To my mind this significantly increases the relevance of the work.

In view of publication in Nature Communications I have the following concerns:

The claims of the title relate to the speed of acquiring isolation between regions covered by liquid electrolyte.

In order for the claims to be corroborated, the paper would need to employ a well argued metric for the speed of the process, and to compare the presented method with existing competition. In order to do this, the time required for the full sequence of fabrication leading up to the resulting isolation should be included. I suggest either changing the claim/title and abstract or to add substantial work to give a basis for the claim of speed.

Furthermore, a thorough analysis of the precision of the localization is expected for this to qualify as a signifying concept for the work. Again, I suggest removing this from the title/abstract or to provide more data. Were there attempts to produce even higher device areal densities? If not, how do we know where the precision limitation of the process lies? What are the expectations for this limitation, and what reasoning would such expectations be based on?

In the paper it is mentioned that "systematically investigating the surface's adhesion is of great importance" in order to describe and explain the process and its potential. I cannot see that such an investigation has been undertaken. There is a comparison of contact angles for a limited selection of surfaces, but this set of data gives no grounds for a model or elaborated explanation regarding the differences observed between surfaces. Which are the motives for choosing the particular surface modification using FAS-17?

The authors argue that it is important for the overall reliability of a system of serially connected devices for the devices to be quite equal in behavior. I do not see how this translates into requiring that the spacing between devices needs to be "ultrasmall" (whatever that means). What should be included is a critical comparison between the electrical behaviors of individual devices (which is lacking) and information on performance as a function not only on the number of cycles, but also as a function of time. The data presented in Figure 4e might contain information regarding the spread of characteristics from device to device, but it is not easy to discern from the way it is presented.

Figure 5 shows results from wireless charging and subsequent discharging. In order for the outcome of charging for different durations to be included as a result, there needs to be some level of explanation of the observed trend. Specifications for power consumption, current level and voltage requirement for the display screen need to be included also.

To summarize:

The noteworthy results consist of the demonstration of a novel method for fabricating dense arrays of micro-supercapacitors which can successfully be connected in series. The performance reported is state-of-the-art and the method for electrolyte separation employed could be useful for many.

However, for the authors claims on precision, speed and reliability to be corroborated more data needs to be presented. In conjunction with presenting the data, the paper needs to include analysis,

interpretation and grounded conclusions based on such data.

Without a substantial boost of supporting data and analysis for at least some of the claims stated, I do not think Nature Communications is the most appropriate journal to publish the work in.

Sincerely,

Per Lundgren
Dept. of Microtechnology and Nanoscience
Chalmers University of Technology

Reviewers' comments:

Reviewer #1:

This manuscript describes a self-splitting mechanism of aqueous electrolytes, enabling the fabrication of compact and customizable micro-supercapacitors. The method of preparing the devices is novel, relying on a unique FAS-17 treated glass surface to tune adhesion properties. This allows precise electrochemical isolation between devices. This approach avoids coating the entire surface, thereby minimizing the risk of contamination to other functional electronic components. The devices are comprehensively characterized for their properties and performance, with particular emphasis on the ability to maintain an exceptionally high areal output voltage of 555 V cm⁻² and a stable capacitance retention of 100% after 9000 cycles, even at a high output voltage of 190 V (fig 4). Moreover, the device exhibits wireless charging capabilities with receiving and transmitting coil system, which could drive a display screen for 30 minutes. This represents a significant advancement in potential microelectronics applications. In summary, the manuscript presents a novel device with significant potential for various microdevices, showcasing precise control of electrolyte localization. However, prior to recommending it for publication, the following questions need to be addressed and discussed.

The authors claim to pattern the electrolyte by a spontaneous droplet self-splitting strategy on a treated surface. However, it needs further intervention by sucking up the excess solution using a pipette to trigger manual or partially guided self-splitting of electrolytes (Supplementary Video 7).

1. It's a bit premature to claim it as a self-splitting strategy please comment on this.

How can the authors precisely control the amount or volume of electrolyte that is removed despite strong hydrogen bonding between the electrolyte droplets? Due to the same reason how do the authors guarantee electrochemical isolation between devices, particularly given the narrow inter-device spacing of 100 μm?

Response: We appreciate the reviewer very much for the positive recommendation and valuable comments. We would like to clarify that our experiment actually consists of simply allowing the electrolyte to flow over a tilted patterned heterogeneous adhesive

surface, during which it gets spontaneously anchored on the highly adhesive regions, achieving spontaneous spatial separation, as observed in Supplementary Video 1, without further intervention. The “sucking up” is only a method we used to observe and record this process under an optical microscope (Supplementary Video 7). Since the glass with patterned surface cannot be tilted under the lens, a pipette is needed to remove excess solution and simulate the process of electrolyte flowing over the microcells. To clarify this possible misunderstanding, **we have re-recorded Supplementary Video 7 and modified “electrolyte self-splitting” to “patterned heterogeneous adhesive surface induce electrolyte directed assembly” in the revised manuscript.**

This strategy can only qualitatively control the amount or volume of electrolyte, depending on the tilt of the patterned surfaces and the speed of electrolyte flow over the microcells¹. We do not precisely control the volume of electrolyte as it does not affect the performance of the individual cell.

Due to the large difference in adhesivity of electrolyte with FAS-17-treated and untreated regions, a certain amount of electrolyte gets anchored on the untreated surface after the electrolyte flows over it while the FAS-17-treated region does not retain any electrolyte. This leads to successful electrochemical isolation, even though inter-cell spacing is narrower than 100 μm . This is seen from the fact that EMImBF₄ electrolyte can assemble onto the rectangle arrays with width of 100 and 50 μm , and inter-cell spacing from 80 to 20 μm (Fig. R1). We have added a clearer discussion with more details in the revised manuscript as follows:

Revision made: *“To understand the mechanism of such precise electrolyte assembly over the patterned heterogeneous surface, we compared the surface adhesion properties of EMImBF₄ electrolyte over the FAS-17-treated and untreated regions. As shown in Figs. 1c-e, the SCAs between EMImBF₄ electrolyte and MIMSCs region (labelled region 1) constituting MXene microelectrodes and interdigital spaces are 83° and 63°, respectively, suggesting greater wettability compared to the FAS-17-treated region (between adjacent cells, labelled region 2 in Fig. 1b) with SCA of 98°. However, the major driving force behind the patterned distribution of the electrolyte is the large*

difference between the adhesivity characteristics of region 1 and 2. The receding contact angle (RCA) between EMImBF₄ electrolyte and region 1 was less than 5°, indicating high adhesion^{2,3}. This explains why electrolyte droplet contact with the MXene film or untreated glass surface strongly adheres to them and is difficult to remove (Supplementary Videos 2-4). However, the RCA was more than 120° for FAS-17-treated glass indicating its low adhesion^{2,3}. Naturally, the electrolyte droplet slips easily without leaving residue due to the substantially low adhesion (Supplementary Video 5). Consequently, when the electrolyte flows over a patterned heterogeneous adhesive surface with large difference in adhesion, it spontaneously gets anchored on the more adhesive region and slides off the less adhesive region (Supplementary Videos 6 and 7), thus achieving precise localization of the electrolyte in the desired area.”

Fig. R1. Patterned heterogeneous adhesive surface inducing EMImBF₄ electrolyte directed assembly onto the rectangle arrays with different dimension design, a fixed microcell width of 100 μm and a changing inter-cell spacing of (a) 80, (b) 50, and (c) 20 μm, and a fixed microcell width of 50 μm and a changing inter-cell spacing of (d) 80, (e) 50, and (f) 20 μm.

2. Moreover, the authors investigated the surface’s adhesion of electrolyte droplet (fig 1c). However, it is important to consider the possibility of electrolyte contact between

devices if the entire system is tilted or moved creating electrochemical short. Comments addressing this concern would be helpful.

Response: We thank the reviewer very much for the suggestive comment. As mentioned by the reviewer, if the entire system is severely tilted or moved, the liquid electrolytes tend to contact between microcells due to their fluidic characteristic, creating electrochemical short circuits. To address this issue, we introduced a small amount of polymer monomer and photo-initiator into the electrolyte. The viscosity of the electrolyte mixture was only slightly changed (Fig. R2a), so it also can anchor on the highly adhesive regions, achieving electrochemical isolation (Fig. R2c). After the exposure to UV irradiation, the electrolyte ink was solidified to form solid-state gel electrolyte directly on the microcell regions (Fig. R2b,d). We will follow this up with more in-depth and systematic research work.

Fig. R2. Optical photographs of (a) the electrolyte mixture and (b) a drop of electrolyte mixture on glass before and after UV irradiation. (c, d) Optical microscopy images of electrolyte mixture anchored on the highly adhesive regions of FAS-17-patterned glass before and after UV irradiation.

3. One aspect that is unclear is the consistency and stability of FAS-17 material patterning, especially during lift-off process of photoresist, without encountering issues

of delamination or peeling-off the FAS-17 layer (fig 1a)?

Response: We thank the reviewer very much for the valuable comment. To verify consistency and stability of FAS-17 material patterning, we performed the time-of-flight secondary-ion mass spectrometry to analyze the surface chemical composition distribution of FAS-17-patterned glass surface after lift-off process in ethanol. As shown in Fig. R3, the top view image of -F fragment represents the distribution of FAS-17. It is clearly seen that FAS-17 has a very uniform distribution over a large area of 3.5 mm × 6.1 mm, which is consistent with the designed pattern before lift-off process of photoresist in ethanol, indicating good consistency and stability of FAS-17 material patterning during the lift-off process.

Fig. R3. Top view time-of-flight secondary-ion mass spectrometry image of -F fragment of FAS-17-patterned glass surface after lift-off process in ethanol.

Additionally, we have also evaluated the stability of FAS-17 layer on glass during the lift-off. Specifically, we soaked FAS-17 treated glass in ethanol for 1 minute, then tested its SCAs with water at different positions, and conducted 10 repeated experiments. The results showed that after 10 soaking cycles, the SCAs remained almost unchanged at about 102° and continued to exhibit strong hydrophobicity (Fig. R4). Therefore, during the photoresist lift-off process, FAS-17 did not encounter any

delamination or peeling-off. In addition, the SCAs at different positions are almost identical, which further proves the consistency of the FAS-17 layer on the glass.

We have discussed the consistency and stability of FAS-17 layers and added Figs. R3,4 in this letter into the revised manuscript.

Fig. R4. The SCAs of water with FAS-17-treated glass at different positions as a function of N, N represents the number of times FAS-17-treated glass soaked in ethanol.

Revision made: “Subsequently, the patterned heterogeneous adhesive surface was obtained by washing away the photoresist layer using ethanol. Time-of-flight secondary-ion mass spectrometry was performed to analyze the distribution of FAS-17. The top view image of -F fragment represents the distribution of FAS-17 (Supplementary Fig. 4). It is clearly seen that FAS-17 has a very uniform distribution over a large area of 3.5 mm × 6.1 mm, which is consistent with the designed pattern before lift-off process of photoresist in ethanol, indicating good stability of FAS-17 patterns during the process. Additionally, the stability and consistency of FAS-17 layer on glass during the photoresist lift-off process was also verified by the unchanged static contact angle (SCA) at different positions during immersion in ethanol for 10 cycles (Supplementary Fig. 5).”

4. The authors charged 72 cells connected in series (arranged in 8 rows) and parallel (9 columns) at 190 V. Consequently, each individual device is minimally charged at over 20 V. This voltage not only raises concerns about high voltage arcing but also may cause gas evolution. It is imperative for the authors to elucidate how the device addresses this issue. Given that the stability window in aqueous environment is only 1.21 V, and with an air break done voltage $\sim 3\text{V}/\mu\text{m}$ and the authors charged the individual MSC only up to 2.5 V (instead of the 20V claimed) (fig 3) in three different ionic liquid-based electrolytes (EMIMBF₄, LiCl and H₂SO₄). Charging the MSC beyond the stability window may cause water splitting reactions and damage the device. The authors should provide information on the stability of device when operating beyond the range of electrochemical water splitting.

Response: We apologize for the misunderstanding. In this work, we have fabricated two types of MIMSCs. One was shown in Fig. 4a, consisting of 72 microcells completely connected in series. This type of MIMSCs can be charged to 190 V in EMIMBF₄ electrolyte, with a charging voltage of 2.64 V distributed to each individual cell, which is within the electrochemical stability window of 2.7 V for the individual MSC (Fig. 3b,c). So, there is no risk of electrolyte decomposition and cell damage. Another type of MIMSCs, as shown in Supplementary Fig.9, consists of 72 microcells connected in series (arranged in 8 rows) and parallel (9 columns). It was only charged to 21 V in EMIMBF₄ electrolyte, with a charging voltage of 2.625 V distributed to each individual cell, which is also within the stability voltage window of 2.7 V for the individual MSC (Fig. 3b,c). To avoid misunderstanding, we have provided a more detailed explanation in the revised manuscript.

5. In considering practical applications beyond wireless charging system, the intrinsic performance of self-discharge is crucial, defining the duration for which the system can function independently without requiring an external charge. The authors charged the array of MIMSCs at a voltage of 190 V, and it would be interesting to investigate the self-discharge behavior after it charges up to 190 V. given such high voltage one would expect strong side reaction, material degradation, water splitting and arc discharge.

Response: We appreciate the reviewer very much for the valuable comment. As

suggested by the reviewer, the self-discharge behavior of MIMSCs after being charged up to 190 V has been investigated and added as Supplementary Fig. 11 in the revised manuscript (Fig. R5 in this letter). Additionally, in this pack of MIMSCs consisting of 72 microcells completely connected in series, no side reactions, material degradation, water splitting and arc discharge occurred as explained above.

Revision made: “The self-discharge time of fully charged MIMSCs from 190 V to 82 V is 1.5 h (Supplementary Fig.11), because the reliable electrochemical isolation effectively prevented the leakage current between individual microcells.”

Fig. R5. Self-discharge profile of MIMSCs consisting of 72 microcells connected in series, obtained immediately after charging to 190 V. The starting voltage of 180 V is caused by switching the testing software.

Reviewer #2:

This is a very interesting report on a novel technique to fabricate micro-supercapacitors in a way that allows larger scale series connections of devices on a smaller footprint area than what has been shown before. The authors convincingly demonstrate the viability of the technique and the performance, in particular in terms of generating a high voltage per footprint area, is impressive.

I appreciate the effort to show performance of the devices using a range of electrolytes.

To my mind this significantly increases the relevance of the work.

In view of publication in *Nature Communications* I have the following concerns:

The claims of the title relate to the speed of acquiring isolation between regions covered by liquid electrolyte.

In order for the claims to be corroborated, the paper would need to employ a well argued metric for the speed of the process, and to compare the presented method with existing competition. In order to do this, the time required for the full sequence of fabrication leading up to the resulting isolation should be included. I suggest either changing the claim/title and abstract or to add substantial work to give a basis for the claim of speed.

Response: We appreciate the reviewer very much for the positive recommendation and valuable comments. As the reviewer pointed out, the claims regarding the speed of electrolyte isolation should be calculated based on the time required for the full sequence of fabrication leading up to the resulting isolation, including lithography process and surface modification with FAS-17. In our manuscript, the time required for the full sequence to deposit electrolyte over 72 and 10000 cells is almost the same, including 8 minutes of lithography process, 3 h of surface modification, and a few seconds of electrolyte deposition. This can be normalized to speeds of 0.0064 and 0.887 cell per second for 72 and 10000 integrated microcells, respectively. In other words, our strategy has a scale effect, and since the time required to deposit electrolyte is independent of the number of integrated MSCs, the speed of final cell production is faster with increasing cell number. The current competitive strategy for adding electrolyte is 3D printing⁴ and electrohydrodynamic jet printing⁵. Although previously reported works do not explicitly mention the speed, it can be analyzed from the supporting videos that the time required for adding gel electrolyte to 13 cells using 3D printing technology is 36 seconds, and the corresponding speed is 0.36 cell per second, and the time required for adding electrolyte to 3 cells using electrohydrodynamic jet printing technology is 20 seconds, the corresponding speed is 0.15 cell per second. Since these techniques must add electrolyte individually to each cell, assuming constant speed of adding electrolytes using these two techniques, the total time required to deposit electrolytes over all cells will increase linearly with the number of cells

prepared. To deposit electrolyte over 10000 cells, it would take roughly 7.7 h and 18.5 h respectively, far exceeding the time (~3.2 h) required by our strategy (Fig. R6). However, considering that we are unable to provide a quantitative description of speed for MIMSCs with different number of cells, to avoid misunderstanding, we accept the reviewer's suggestion to modify the title and abstract in our revised manuscript.

Revision made: *We have revised the title of our revised manuscript to “Monolithically integrated micro-supercapacitors with high areal number density produced by controlling electrolyte directed assembly”.*

Fig. R6. The time to deposit electrolyte over 10000 cells required by our strategy compared with the state-of-the-art methods reported^{4,5}.

2. Furthermore, a thorough analysis of the precision of the localization is expected for this to qualify as a signifying concept for the work. Again, I suggest removing this from the title/abstract or to provide more data. Were there attempts to produce even higher device areal densities? If not, how do we know where the precision limitation of the process lies? What are the expectations for this limitation, and what reasoning would such expectations be based on?

Response: We appreciate the reviewer very much for the suggestive comments. As is well known, reducing the size of individual cell and inter-cell spacing are essential to achieve a high areal integration density in the MIMSCs. As suggested by the reviewer, we have attempted to produce MIMSCs with higher number density per unit area by

fixing the inter-cell distance to 100 μm , changing the microcell width from 500 μm to 50 μm (Figs. R7,8), with the aim of exploring the precision limitation of our strategy. As shown in Fig. R7, microelectrode arrays with microcell width up to 200 μm were produced and electrolytes were successfully anchored onto microcell arrays by our strategy. For example, the 3×3 MIMMSCs, in which each MSC unit has an electrode length of 200 μm , width of 75 μm , gap of 50 μm and an inter-cell spacing of 100 μm , achieved an ultra-high areal cell number density of 1404 cell cm^{-2} . When the width of microcells was decreased to 100 and 50 μm , the electrode widths were designed to be 25 μm and 20 μm . These widths were too small, causing the electrode material to detach from the current collector during the lift-off process in ethanol, resulting in the failure of microelectrode arrays preparation, as shown in Fig. R8. Furthermore, we continued exploring the precision limitation of patterned heterogeneous adhesive surface-induced electrolyte directed assembly strategy by fixing the microcell width to 100 and 50 μm , changing the inter-cell spacing from 100 to 20 μm . As shown in Fig. R9, electrolytes could be precisely assembled on all designed square arrays, achieving an ultra-high areal cell number density of 24930 cell cm^{-2} . We have added Fig. R7 in this letter as Supplementary Fig. 14 in the revised manuscript.

It should be noted that even higher areal cell number can be achieved by using more efficient lithography equipment with higher precision. We will conduct more systematic research on this part in our future work.

Revision made: *“We also produced MIMMSCs with higher number density per unit area by fixing the inter-cell spacing to 100 μm , changing the microcell width from 500 to 200 μm , with the aim of exploring the precision limitation of our strategy. As shown in Supplementary Fig. 14, microelectrode arrays with microcell width up to 200 μm were produced and electrolytes were successfully anchored onto microcell arrays by our strategy. For example, the 3×3 MIMMSCs, in which each MSC unit has an electrode length of 200 μm , width of 75 μm , gap of 50 μm and an inter-cell spacing of 100 μm , achieved an ultra-high areal cell number of 1404 cell cm^{-2} .”*

Fig. R7. Microscope images of MIMSCs with a fixed inter-cell spacing ($100\ \mu\text{m}$) with microcell widths of 500 , 400 , 300 and $200\ \mu\text{m}$, microelectrode arrays without electrolyte (a-d) and with EMImBF₄ electrolyte (e-h).

Fig. R8. Microscope images of MIMSCs with a fixed inter-cell spacing ($100\ \mu\text{m}$) and microcell widths of $100\ \mu\text{m}$ (a) and $50\ \mu\text{m}$ (b).

Fig. R9. Patterned heterogeneous adhesive surface-induced electrolyte directed assembly onto the rectangle arrays with different dimension design, a fixed microcell width (100 μm) with inter-cell spacings of 100 μm (a), 80 μm (b), 50 μm (c) and 20 μm (d), a fixed unit cell width (50 μm) with inter-cell spacings of 100 μm (e), 80 μm (f), 50 μm (g) and 20 μm (h).

3. In the paper it is mentioned that “systematically investigating the surface’s adhesion is of great importance” in order to describe and explain the process and its potential. I cannot see that such an investigation has been undertaken. There is a comparison of contact angles for a limited selection of surfaces, but this set of data gives no grounds for a model or elaborated explanation regarding the differences observed between surfaces. Which are the motives for choosing the particular surface modification using FAS-17?

Response: We appreciate the reviewer very much for the valuable comments. In our future work, more systematic investigation of the contact angles involving various electrolytes and new electrode materials would be necessary to ascertain the wide generalization and applicability of our method. It is noted that this study will fall outside the scope of this current work. Therefore, we have removed the phrase “systematically investigating the surface’s adhesion is of great importance”. In this work, we mainly focus on the difference between EMIMBF₄’s electrolyte’s adhesion on three regions of interest-the surface of MXene microelectrodes, the interdigital region between the

microelectrodes and the FAS-17-treated region between adjacent cells, to explain why the electrolyte gets precisely assembled over the desired active area. As suggested by the reviewer, we have clarified the discussion about the information that can be gleaned from the static contact angle (SCA) and receding contact angle (RCA) between the liquid and surfaces of interest, as follows:

Revision made: *“To understand the mechanism of such precise electrolyte assembly over the patterned heterogeneous surface, we compared the surface adhesion properties of EMImBF₄ electrolyte over the FAS-17-treated and untreated regions. As shown in Figs. 1c-e, the SCAs between EMImBF₄ electrolyte and MIMSCs region (labelled region 1) constituting MXene microelectrodes and interdigital spaces are 83° and 63°, respectively, suggesting greater wettability compared to the FAS-17-treated region (between adjacent cells, labelled region 2 in Fig. 1b) with SCA of 98°. However, the major driving force behind the patterned distribution of the electrolyte is the large difference between the adhesivity characteristics of region 1 and 2. The receding contact angle (RCA) between EMImBF₄ electrolyte and region 1 was less than 5°, indicating high adhesion^{2,3}. This explains why electrolyte droplet contact with the MXene film or untreated glass surface strongly adheres to them and is difficult to remove (Supplementary Videos 2-4). However, the RCA was more than 120° for FAS-17-treated glass indicating its low adhesion^{2,3}. Naturally, the electrolyte droplet slips easily without leaving residue due to the substantially low adhesion (Supplementary Video 5). Consequently, when the electrolyte flows over a patterned heterogeneous adhesive surface with large difference in adhesion, it spontaneously gets anchored on the more adhesive region and slides off the less adhesive region (Supplementary Videos 6 and 7), thus achieving precise localization of the electrolyte in the desired area.”*

Fig. R10. The SCAs and RCAs of EMImBF₄ electrolyte with MXene microelectrodes (a), interelectrode glass (b), and treated inter-cell glass (c).

Based on the experimental design, we considered the following aspects when choosing the surface modification of FAS-17: Firstly, FAS-17 is a widely used surface modifier that can achieve surface hydrophobicity and low adhesion treatment^{3,6,7}. Secondly, FAS-17 combines the characteristics of organic silicon and organic fluorine. After hydrolysis, the silane functional group can react with the silicon hydroxyl group on the glass surface, producing a strong binding force, and thus providing the stabilization during the subsequent lift-off process in ethanol without damage (Figs. R3,4 in this letter). Thirdly, FAS-17 can achieve surface modification under a relatively mild condition (chemical vapor deposition at 70 °C for 3 h) with minimal damage to electrode materials.

4. The authors argue that it is important for the overall reliability of a system of serially connected devices for the devices to be quite equal in behavior. I do not see how this translates into requiring that the spacing between devices needs to be “ultrasmall” (whatever that means). What should be included is a critical comparison between the electrical behaviors of individual devices (which is lacking) and information on performance as a function not only on the number of cycles, but also as a function of

time. The data presented in Figure 4e might contain information regarding the spread of characteristics from device to device, but it is not easy to discern from the way it is presented.

Response: We thank the reviewer very much for raising this critical concern. Integration of MSCs is important for broadening the capacitance range and operational voltage for on-demand customization of micro-electronic products. So far, scalable high-density integration with unit consistency remains challenging. Intuitively, reducing the size of unit cell and space between microcells are essential to achieve high areal integration density and miniaturization in the MIMSCs. We would like to clarify that, “ultrasmall space between microcells” is not the requirement for the reliability the MIMSCs but for high-density integration and miniaturization of the MIMSCs. In order to avoid the misunderstandings about “ultrasmall space between microcells”, we have made revisions to section 2.3 in the revised manuscript. Additionally, as suggested by the reviewer, we have compared between the electrical behaviors of individual cells (Fig. R11e) and adjusted the information on performance as a function not only on the number of cycles, but also as a function of time (Fig. R11h, Fig. 4 in revised manuscript).

Revision made: *“Besides employing electrolytes to regulate the output voltage/capacitance, another more straightforward approach is to integrate multiple MSC units in series and/or parallel, thus generating a superposition of output voltage/capacitance⁸. So far, constructing MIMSCs with high-density and stable performance has remained challenging. To demonstrate the advantages of our strategy, MIMSCs comprising 72 cells with customized serial and/or parallel configurations, were integrated on a small substrate (0.58 cm × 0.59 cm, Fig. 4a and Supplementary Fig. 12a). The interelectrode spacing of the microcell was 50 μm, and the inter-cell spacing was only 100 μm, achieving a high space utilization and reaching an excellent areal number density of 210 cells cm⁻², far superior to the previously reported integrated MSCs, with number density lower than 158 cells cm⁻² (Fig. 4h)⁹⁻¹². The successful ability to compactly integrate fully functioning MSCs is attributed to the spatial control of electrolyte (Supplementary Fig. 9) by constructing patterned heterogeneous adhesive surface on the entire MIMSCs areas. Moreover, even a slight*

discrepancy between individual microcells may lead to a big difference in voltage division, cascading into uneven energy distribution throughout the monolithic system. This will result in some units not being fully activated while others are overworked, eventually causing the MIMSCs to fail unpredictably. Thus, the performance consistency among all the microcells is imperative to the successful integration. The three-dimensional thickness mapping of a random region and the height profiles of eight individual MSCs in the MIMSCs showing the consistency of microcells' thickness, indicate that the microfabrication process was highly controllable over a large scale (Fig. 4b,c and Supplementary Fig.10). Benefiting from the thickness consistency and accurate coverage of electrolyte on microcells, the capacitance difference between all the microcells is limited to within 10% (Fig. 4d), which ensures the monotonously increasing voltage to 190 V of MIMSCs with 72 serially-connected microcells (Fig. 4e) in EMIMBF₄ electrolyte. Additionally, the correlation coefficient calculated from the plot of response capacitance versus cell number was close to 1, demonstrating the excellent modularity and performance uniformity of all the on-chip MIMSCs (Fig. 4e). Besides, our MIMSCs could operate under 190V stably with acceptable coulombic efficiency at different current densities (Fig. 4f), ultimately delivering a high systemic energy density of 14.8 mWh cm⁻³ (0.37 μWh cm⁻²) based on the whole model including inter-cell spacing. The self-discharge time of fully charged MIMSCs from 190 V to 82 V is 1.5 h (Supplementary Fig.11), because the reliable electrochemical isolation effectively prevented the leakage current between individual microcells. This pack of MIMSCs demonstrated almost 100% capacitance retention after 9000 continuous long-term cycles at an applied voltage of 190 V (Fig. 4g), demonstrative of exceptional performance consistency among all the microcells. It should be noted that, the value of 190 V is one of the highest operating voltages ever reported^{10,12-15}.”

Fig. R11. Electrochemical characterization of MIMSCs with 72 serially-connected microcells. **a**, Photograph. **b,c**, Three-dimensional thickness mapping of four selected MSCs (**b**) and height profiles of randomly selected MSCs (**c**) at different locations in MIMSCs model. **d**, Capacitance obtained at 1 μA of each MSC unit within the MIMSCs. **e**, Total voltage/capacitance response versus cell number of MIMSCs containing 8, 16, 24, 32, 40, 56, and 72 serially-connected microcells, the capacitance was calculated from GCD profiles measured at 2 μA. **f,g**, GCD profiles obtained at currents of 0.2-5 μA (**f**) and long-term cycling stability (**g**) of 72 serially-connected microcells under 190 V. Inset in (**g**) shows the initial and last 3 GCD profiles during the 9000-cycle run. **h**, Areal cell number and areal output voltage of our MIMSCs compared with previous reports^{5,6,14,13,27,33}.

5. Figure 5 shows results from wireless charging and subsequent discharging. In order for the outcome of charging for different durations to be included as a result, there needs to be some level of explanation of the observed trend. Specifications for power

consumption, current level and voltage requirement for the display screen need to be included also.

Response: We thank the reviewer very much for the valuable comments. As suggested, we have provided a reasonable explanation of the observed trends of discharging time when charging for different durations in our revised manuscript as below.

Revision made: “Upon removing the transmitting coil, the MIMSCs could steadily discharge over a long duration at 1 μ A (Fig. 5d). According to the discharge profiles and capacitance of MIMSCs obtained after wireless charging for different periods, it can be seen that only wireless charging of 2 seconds, the MIMSCs achieved 88% charging capacitance. Meanwhile, the voltage reached 5.4 V, and the current of wireless charging rapidly decreased^{16,17}. The MIMSCs continued to be charged under a low current and the capacitance maintained almost unchanged after wireless charging of 20 seconds, validating the applicability of our MIMSCs.”

As suggested, the specifications for power consumption have been supplemented in the revised manuscript.

Revision made: “The driving voltage is 3 V and the current is less than 5 μ A of the display screen used in this study.”

To summarize:

The noteworthy results consist of the demonstration of a novel method for fabricating dense arrays of micro-supercapacitors which can successfully be connected in series. The performance reported is state-of-the-art and the method for electrolyte separation employed could be useful for many.

However, for the authors claims on precision, speed and reliability to be corroborated more data needs to be presented. In conjunction with presenting the data, the paper needs to include analysis, interpretation and grounded conclusions based on such data. Without a substantial boost of supporting data and analysis for at least some of the claims stated, I do not think *Nature Communications* is the most appropriate journal to publish the work in.

Response: We thank the reviewer very much for the valuable comments. We have

addressed your comments above with detailed point-to-point answers and indications of where changes have been introduced. All major changes have been highlighted in **blue** in the revised manuscript and Supplemental Information (SI). We believe the current version of manuscript would be suitable for *Nature Communications*.

References in this letter

1. Yuan, S., et al. Fabrication of flexible and transparent metal mesh electrodes using surface energy-directed assembly process for touch screen panels and heaters. *Adv. Sci.* **10**, (2023).
2. Yin, K., et al. Femtosecond laser thermal accumulation-triggered micro-/nanostructures with patternable and controllable wettability towards liquid manipulating. *Nanomicro Lett.* **14**, 97 (2022).
3. Li, H., et al. Droplet precise self-splitting on patterned adhesive surfaces for simultaneous multidetection. *Angew. Chem. Int. Ed.* **59**, 10535-10539 (2020).
4. Wang, S., et al. Monolithic integrated micro-supercapacitors with ultra-high systemic volumetric performance and areal output voltage. *Natl. Sci. Rev.* **10**, nwac271 (2023).
5. Lee, K.-H., et al. Ultrahigh areal number density solid-state on-chip microsupercapacitors via electrohydrodynamic jet printing. *Sci. Adv.* **6**, eaaz1692 (2020).
6. Wan, X., et al. A Wetting-enabled-transfer (WET) strategy for precise surface patterning of organohydrogels. *Adv. Mater.* **33**, e2008557 (2021).
7. Sun, J., et al. Patterning a superhydrophobic area on a facile fabricated superhydrophilic layer based on an inkjet-printed water-soluble polymer template. *Langmuir* **36**, 9952-9959 (2020).
8. Fan, Z., et al. Towards kilohertz electrochemical capacitors for filtering and pulse energy harvesting. *Nano Energy* **39**, 306-320 (2017).
9. Ma, J. X., et al. Aqueous MXene/PH1000 hybrid inks for inkjet-printing micro-supercapacitors with unprecedented volumetric capacitance and modular self-powered microelectronics. *Adv. Energy Mater.* **11**, 2100746 (2021).
10. Shi, X., et al. Ultrahigh-voltage integrated micro-supercapacitors with designable shapes and superior flexibility. *Energ. Environ. Sci.* **12**, 1534-1541 (2019).
11. Kamboj, N., et al. Ultralong cycle life and outstanding capacitive performance of a 10.8 V metal free micro-supercapacitor with highly conducting and robust laser-irradiated graphene for an integrated storage device. *Energ. Environ. Sci.* **12**, 2507-2517 (2019).
12. Li, X., et al. High-voltage flexible microsupercapacitors based on laser-induced graphene. *ACS Appl. Mater. Inter.* **10**, 26357-26364 (2018).
13. Han, F., et al. Structurally integrated 3D carbon tube grid-based high-performance filter capacitor. *Science* **377**, 1004-1007 (2022).

14. Wu, M., et al. Arbitrary waveform AC line filtering applicable to hundreds of volts based on aqueous electrochemical capacitors. *Nat. Commun.* **10**, 2855 (2019).
15. Bai, S., et al. High voltage microsupercapacitors fabricated and assembled by laser carving. *ACS Appl. Mater. Inter.* **12**, 45541-45548 (2020).
16. Gao, C., et al. A seamlessly integrated device of micro-supercapacitor and wireless charging with ultrahigh energy density and capacitance. *Nat. Commun.* **12**, 2647 (2021).
17. Xu, S., et al. Stretchable batteries with self-similar serpentine interconnects and integrated wireless recharging systems. *Nat. Commun.* **4**, 1543 (2013).

REVIEWERS' COMMENTS

Reviewer #1 (Remarks to the Author):

Thank you for your detailed response to my comments and for the additional experiments conducted to clarify the aspects I raised regarding your manuscript titled "Monolithically integrated micro-supercapacitors with high areal number density produced by controlling electrolyte directed assembly."

Your efforts in re-recording supplementary videos and revising the manuscript to clarify the "patterned heterogeneous adhesive surface induce electrolyte directed assembly" process are commendable. The modifications and additional details provided have significantly enhanced the clarity and depth of your study.

The innovative approach to achieving spontaneous spatial separation and the subsequent clarification on the electrolyte assembly process are particularly noteworthy. Your strategy to address potential electrochemical shorts by incorporating a polymer monomer and photo-initiator into the electrolyte, thus enabling electrochemical isolation, is a testament to the thoroughness of your research.

Moreover, your investigation into the adhesion properties of electrolyte droplets and the stability of the FAS-17 layer adds valuable insights into the fabrication process and the functionality of the micro-supercapacitors. The systematic research work proposed to further explore these aspects is indeed a step in the right direction.

Given the comprehensive revisions and the additional experiments that justify the claims made, along with the technology's potential to address one of the challenges in micro-energy storage devices, namely packaging, I believe the paper now reads well and can be accepted for publication.

The advancements documented in your study could significantly contribute to solving packaging problems and enhancing the sustainability of micro-energy storage devices. I look forward to seeing this valuable work published and to its impact on the field.

Reviewer #2 (Remarks to the Author):

The authors have made a serious effort to address all the comments I have given on the manuscript, and there has been a significant improvement in its suitability for publication in Nature Communications.

The remaining comments/questions I have are:

1. The variation in capacitance shown in Fig. 4 d is claimed to be within 10%. I strongly recommend explicitly stating the actual variation in values, e. g. 1.5 - 1.8 microfarads - if that is the variation - the graph in Fig. 4 d is not at all easy to read.
2. The following sentences have grammatical errors or are poorly constructed: first sentence of Abstract, first sentence of Conclusion.
3. The title "Monolithically integrated micro-supercapacitors with high areal number density produced by controlling electrolyte directed assembly" is a bit awkward and I recommend rephrasing "...by controlling electrolyte directed assembly" maybe to something like "...by surface adhesion controlled electrolyte assembly"

Reviewers' comments:**Reviewer #1:**

Thank you for your detailed response to my comments and for the additional experiments conducted to clarify the aspects I raised regarding your manuscript titled "Monolithically integrated micro-supercapacitors with high areal number density produced by controlling electrolyte directed assembly."

Your efforts in re-recording supplementary videos and revising the manuscript to clarify the "patterned heterogeneous adhesive surface induce electrolyte directed assembly" process are commendable. The modifications and additional details provided have significantly enhanced the clarity and depth of your study.

The innovative approach to achieving spontaneous spatial separation and the subsequent clarification on the electrolyte assembly process are particularly noteworthy. Your strategy to address potential electrochemical shorts by incorporating a polymer monomer and photo-initiator into the electrolyte, thus enabling electrochemical isolation, is a testament to the thoroughness of your research.

Moreover, your investigation into the adhesion properties of electrolyte droplets and the stability of the FAS-17 layer adds valuable insights into the fabrication process and the functionality of the micro-supercapacitors. The systematic research work proposed to further explore these aspects is indeed a step in the right direction.

Given the comprehensive revisions and the additional experiments that justify the claims made, along with the technology's potential to address one of the challenges in micro-energy storage devices, namely packaging, I believe the paper now reads well and can be accepted for publication.

The advancements documented in your study could significantly contribute to solving packaging problems and enhancing the sustainability of micro-energy storage devices.

I look forward to seeing this valuable work published and to its impact on the field.

Response: We thank the reviewer very much for the positive recommendation.

Reviewer #2:

The authors have made a serious effort to address all the comments I have given on the

manuscript, and there has been a significant improvement in its suitability for publication in *Nature Communications*.

The remaining comments/questions I have are:

1. The variation in capacitance shown in Fig. 4 d is claimed to be within 10%. I strongly recommend explicitly stating the actual variation in values, e. g. 1.5 - 1.8 microfarads - if that is the variation - the graph in Fig. 4 d is not at all easy to read.

Response: We thank the reviewer very much for the positive and valuable comments. As suggested, we changed the description of this part in the revised manuscript.

Revision made: *“Benefiting from the thickness consistency and accurate coverage of electrolyte on microcells, the capacitance of all the microcells varies from 1.55 to 1.77 μ F (Fig. 4d), which ensures the monotonously increasing voltage to 190 V of MIMSCs with 72 serially-connected microcells in EMIMBF₄ electrolyte (Fig. 4e).”*

2. The following sentences have grammatical errors or are poorly constructed: first sentence of Abstract, first sentence of Conclusion.

Response: We thank the reviewer very much for the valuable comment. As suggested, we have revised the first sentence of the Abstract and first sentence of the Discussion in the revised manuscript.

Revision made: *“Accurately placing very small amounts of electrolyte on tiny micro-supercapacitors (MSCs) arrays in close-proximity is a major challenge. This difficulty hinders the development of densely-compact monolithically integrated MSCs (MIMSCs).”*

Revision made: *“In summary, we have successfully tackled a major obstacle in creating ultra-dense MSCs by employing a new method. This method involves a patterned adhesive surface that induces direct assembly of electrolyte into 10000 tiny droplets within seconds.”*

3. The title "Monolithically integrated micro-supercapacitors with high areal number density produced by controlling electrolyte directed assembly" is a bit awkward and I recommend rephrasing "...by controlling electrolyte directed assembly" maybe to something like "...by surface adhesion controlled electrolyte assembly"

Response: We thank the reviewer very much for the valuable comment. As suggested,

we have revised the title as “Monolithically integrated micro-supercapacitors with high areal number density produced by surface adhesive-directed electrolyte assembly” in the revised manuscript.